# Expression Analysis of *IZUMO1* Gene during Testicular Development of Datong Yak (*Bos Grunniens*)

**DOI:** 10.3390/ani9060292

**Published:** 2019-05-29

**Authors:** Qudratullah Kalwar, Xuezhi Ding, Anum Ali Ahmad, Min Chu, Xiaoyun Wu, Pengjia Bao, Ping Yan

**Affiliations:** 1Key Laboratory of Yak Breeding Engineering, Lanzhou Institute of Husbandry and Pharmaceutical Sciences, Chinese Academy of Agricultural Science, Lanzhou 730050, Gansu, China; dingxuezhi@caas.cn (X.D.); anum2017@lzu.edu.cn (A.A.A.); chumin@caas.cn (M.C.); wuxiaoyun@caas.cn (X.W.); baopengjia@caas.cn (P.B.); 2Shaheed Benazir Bhutto University of Veterinary and Animal Sciences, Sakrand 67210, Sindh, Pakistan; 3State Key Laboratory of Grassland Agro-Ecosystems, School of Life Sciences, Lanzhou University, Lanzhou 730050, Gansu, China

**Keywords:** Datong yak, fertilization, IZUMO1, testis, testicular development

## Abstract

**Simple Summary:**

IZUMO1 (IZUMO sperm-egg fusion) is a crucial member of the immunoglobulin superfamily, and it plays a vital part in egg sperm interaction, fusion, and in spermatogenesis. Previous studies identified the fundamental role of the *IZUMO1* gene in meiosis of bovine spermatogenesis and spermatogenic cell development and fertility. The current study confirmed the significant expression of the *IZUMO1* gene with testicular development and its potential role in fertilization of Datong yak. This research explained the biological role of *IZUMO1* in yak reproduction; therefore in the future it might provide a novel prospect to apply data on reproduction and to understand the further regulatory mechanisms of the *IZUMO1* gene during spermatogenesis of different mammalian species.

**Abstract:**

The *IZUMO1* gene has promising benefits for the national development of novel non-hormonal contraceptives and in the treatments of fertility. Understanding the function of *IZUMO1,* its mRNA, and protein expression is critical to gain insight into spermatogenesis and promote sperm-egg fusion during reproduction of Datong yak. Therefore, we estimated the *IZUMO1* gene expression in different ages of Datong yak by using semi quantitative PCR, qPCR, and western blotting. The results of the qPCR, semi-quantitative PCR and western blotting revealed that the expression level of IZUMO1 mRNA was significantly (*p* < 0.05) higher in the testis of 30 months and 6 years old followed by 18 and 6 months old Datong yak, respectively. We also predicted secondary and tertiary protein structure of IZUMO1 by using bioinformatics software that the revealed presence of a signal peptide, Izumo domain, immunoglobulin (Ig) like domain, and transmembrane region. Moreover, immunostaining analysis also elucidated that IZUMO1 was more prominent in the testis of 30 months and 6 years old yak, which represented that the *IZUMO1* gene expression might be higher during the peak breeding ages (6 to 7 years) of the yak, and play a potential role in spermatogenesis, fertility, and testicular development.

## 1. Introduction

Yak (*Bos grunniens*) are one of the most major animals economically and they were domesticated in the mountains of Asia. Yak produce a high quality of meat and are broadly utilized in the transhumance practice system in the severe environment of the alpine regions [1,2] Over the past decades, China began to practice artificial insemination by using frozen semen samples of wild yak and selected within and among the yak breeds to improve the overall performance of the yak breeds. Correspondingly, cattle-yak have been produced by using crossbreeding of exotic cattle and yak, which are mainly beneficial in the aspects of more substantial quantity of milk production, bigger size, stronger labor-force, and having a better heat tolerance [3,4]. Male yaks around the age of six months start to display mounting behavior, but until two years of age, sperm has not been found in the ejaculates of male yak. However, at the age of 3 to 4 years, bulls usually start to mate; and they reach their peak breeding age around 6 to 7 years and at that time they have to start asserting a dominant position in the mating hierarchy of the herd [5].

The Izumo sperm egg fusion gene (*IZUMO1*) is a crucial member of the immunoglobulin superfamily and plays a crucial role in spermatogenesis, sperm egg interaction, and fusion [6,7,8,9]. As reported by Kim [7] the expression of *IZUMO1* mRNA was mainly in the testis and sperm and performed an important part in fertilization. On the other hand, sterile males were reproduced when *IZUMO1* gene was knock-out due to the incapability of sperm to fuse the egg membrane of zona pellucida [10]. A past study also identified that the *IZUMO1* gene played a main role in meiosis of bovine spermatogenesis and spermatogenic cell development and fertility [11]. Moreover, these findings demonstrated that *IZUMO1* is important for sperm oocyte fusion on the sperm side and that it interacts with other proteins either on the oocyte membrane or on the sperm membrane [12]. In yak CNV (copy number variation) of the *IZUMO1* gene was associated with the age of adolescence and sperm production [13]. Yaks are endemic species and they can survive in any harsh environment, high altitude, cold temperature, and in thin air, but yaks generally have a low fertility rate as compared to cattle. Therefore, it is necessary to understand the basic molecular and biological events involved during testicular development of yak. Hence, we estimated the *IZUMO1* gene expression in different ages of Datong yak by using semi quantitative PCR, qPCR, and western blotting, and also predicted IZUMO1 protein structure by utilizing bioinformatics methods. Thus, the general objectives of this study were to look at the potential function of bovine *IZUMO1* gene in fertilization of yak, which can provide an insight into the application of *IZUMO1* as underlying markers in the yak reproduction program. Therefore, understanding its expression and specific gene functions in domestic animals may be highly relevant.

## 2. Materials and Methods

The current study was conducted at the key Laboratory of yak breeding and engineering, Lanzhou Institute of Husbandry and Pharmaceutical Sciences, sample collections were collected in accordance with the guidelines for the care and use of laboratory animals, Lanzhou Institute of Husbandry and Pharmaceutical Sciences, China. Moreover, every necessary effort was made to reduce pain, and each animal was slaughtered under anesthesia. Therefore, we were granted permission to complete this study and the legal certificate number was SCXK (Gan) 2014-0002. The animals were divided into 4 different developmental stages of 6 months, 18 months, 30 months, and 6 years. A total of 16 animals were used in this study. Each age group contained randomly selected 4 male yaks. The testis, liver, kidney, lung, heart, spleen, and intramuscular fat tissues were collected from different ages of the Datong yak breeds after slaughtering. The tissues after extraction were snap frozen in liquid nitrogen for transport to the laboratory and finally kept at −80 °C until further analysis.

This research was conducted between 2018 and 2019 by Prof Ping Yan and QudratullahKalwar at the Key Laboratory of Yak Breeding Engineering, Lanzhou Institute of Husbandry and Pharmaceutical Sciences, Chinese academy of agricultural Sciences, Lanzhou 730050, Gansu, China. All the samples were collected from the Gansu breeding cooperatives and from Qinghai province, China. Sample collection was performed in strict accordance with the guide for the Care and Use of Laboratory Animals, Lanzhou Institute of Husbandry and Pharmaceutical Sciences, China. Additionally, all the animals were slaughtered under anesthesia, and all necessary efforts were made to minimize the risk of suffering. Thus, we agreed to perform research on the yak. The legal certificate number is SCXK (Gan)2014-0002.

### 2.1. Purification of Total RNA 

The Trizol reagent (Tri Pure Isolation Reagent, Roche, USA) was used for complete RNA purification from the tissues, according to the manufacturers’ instructions. The cDNA was prepared from 1µL of RNA by using the PrimeScript^TM^ RT reagent kit (Perfect Real Time) (TaKaRa Bio Inc., Shiga, Japan). Furthermore, PrimeScript^TM^ 1st strand cDNA synthesis kit was also used for the synthesis of cDNA for the cloning of the bovine *IZUMO1* sequence.

### 2.2. Primer Designing and PCR Amplification

The NCBI (National Center for Biotechnology Information) database was used to get bovine gene sequence for designing the primers, which were used for PCR and qPCR (Table 1). The amplification was performed using the qPCR in an overall volume of 25 μL, comprising 10 μM primers, GoTaq^®^ Green Master Mix, 500 ng cDNA, (Promega, Madison, WI, USA). The PCR reactions were carried out as initiation at 95 °C for 2 min, followed by 35 cycles of denaturation for 1 min at 95 °C, annealing for 1 min at 55 to 60 °C, extension at 72 °C for 1 min, final extension at 72 °C for 5 min, and storage at 4 °C. The non-denaturing TBE (Tris Borate EDTA) polyacrylamide gel or 1% agarose gels were used for the visualization of the PCR amplified products. The amplification of the genes was also analyzed by observing the melting curves obtained through the qPCR assay.

### 2.3. Cloning

50 µL polymerase chain reaction was performed to obtain amplified product for the purpose of cloning. The PCR reaction mixture contained one shot LA PCR^TM^ Mix (ver.2.0.) 25 µL, cDNA template 1 µL, 1µL (forward and reverse primers) and sterilized purified water of 28 µL (Takara bio Inc., Shiga, Japan). The PCR reactions profile was as follows; 94 °C for 1 min; followed by 35 cycles of denaturation for 10 s at 94 °C, annealing for 15 min at 61 °C, an extension for 90 s at 72 °C, and a final extension for 15 min at 72 °C. The gel extraction kit (TIANGEN, China) was used for the purification of bands from 1% agarose gel, used for the separation of reaction products, and the purified product was cloned into the pMD19-T vector and sequenced by GENEWIZ, China. Finally the different bio informatics tools were used for sequence analysis.

### 2.4. Quantitative Real-Time PCR

The relative expression levels of the *IZUMO1* gene was determined using qPCR. The bovine GAPDH was used for the reference gene [14]. The qPCR reaction mixture contained 12.5 µL TB Green^TM^ Premix Ex Taq^TM^ II (2X) (Tli RNase H Plus, TaKaRa Bio Inc., Shiga, Japan), 1 µL of 10 mM primers (forward and reverse each), cDNA 2 µL, and purified water 8.5 µL. The polymerase chain reactions were carried out by using the Bio Rad CFX96 Real Time Detection System (Bio Rad, Hercules, CA, USA). The cyclic conditions were as follows: 95 °C for 1 min followed by 39 cycles of denaturation for 10 s at 95 °C, annealing for 30 s at 60 °C and extension for 10 s at 68 °C.

### 2.5. Semi Quantitative PCR

The semi quantitative PCR reactions for exploration of GAPDH and IZUMO1 in yak testis contained a total of 162.5 µL Taq PCR Master Mix (TIANGEN, China), 13 µL of 10 mM primers (forward and reverse each), and 123.5 µL sterilized water. After that, the reaction mixture was split into 24 µL aliquots into 11 tubes, and 1 µL of cDNA templates was mixed into the 10 tubes, and 1 tube was used as a control. The PCR cyclic conditions was; initiation for 5 min at 95 °C, followed by 35 cycles of denaturation for 30 s at 95 °C, annealing for 30 s at 55–60 °C, extension for 20 s at 72 °C; and the last extension for 5 min at 72 °C. Agarose gel (1%) was used for evaluation of reaction products by using ethidium bromide staining.

### 2.6. Bioinformatics Analysis

The ORF (open reading frame) finder program (http://www.ncbi.nlm.nih.gov/gorf/gorf.html) was used to estimate the coding region and the sequence of amino acids of IZUMO1. The conserved domain in IZUMO1 was predicted by Pfam (http://pfam.janelia.org/). The 3D and secondary structures for the protein encoded by IZUMO1 were analyzed via SWISS-MODEL (http://swissmodel.expasy.org/), PDB viewer and PSI pred (http://bioinf.cs.ucl.ac.uk/psipred).

### 2.7. Western Blot of IZUMO1

The proteins were extracted from the testis samples similarly as described by reference [2,15]. Initially proteins were fixed in 12% Tricine SDS-PAGE for polyacrylamide gels then shifted onto PVDF (polyvinylidene difluoride) membranes (Roche, USA), 5% milk powder was used for blocking of the membranes in 1 × 9 PBS and 0.1% Tween-20 for 60 minutes, then washing was done with PBS/Tween and membranes were incubated with anti β-actin (1:1000; Abcam, USA) and anti IZUMO1 antibody (1:1000, Abcam, USA) at 4 °C overnight. The membranes were then incubated with a secondary antibody protein find goat anti rabbit IgG (H + L), HRP (Horseradish peroxidase) conjugate (1:5000; Trans Gen Biotech, Beijing) for 1 h, and ECL (Electrochemiluminescence) detection system (Pierce, USA) was used for visualization of the bands, the protein levels were measured by using densito metric analysis and finally the pictures were taken by X-ray film.

### 2.8. Immuno-Staining of Testis

Deparaffinized sections were autoclaved in a sodium citrate solution to retrieve antigenicity and slides were blocked for one hour at room temperature in 5% serum in PBS and incubated overnight at 4 °C with an Anti-IZUMO1 antibody (1:500; Abcam, UK). Consequently, sections were stained with an HRP conjugated anti rabbit secondary antibody (1:10000; Jackson, USA) and finally a microscope (Leica, UK) was used for capturing the images.

### 2.9. Data Analyses

A quantitative mRNA expression level of the target gene was determined using the threshold cycle 2^−ΔΔCt^ method [15]. Data were expressed as the mean ± SD. The data were analyzed through ANOVA and significant differences were considered at *p* < 0.05.

## 3. Results

### 3.1. The Expression Profile of IZUMO1 Gene by Semi-Quantitative PCR

Primarily, we analyzed the expression level of *IZUMO1* mRNA in 6, 18, 30 months, and 6 years of Datong yak testis, liver, kidney, lung, heart, spleen, and intramuscular fat tissues using semi-quantitative PCR and qPCR (Figure 1 and Figure 2). The results indicated that the *IZUMO1* gene highly expressed in the testis of 30 months and 6 years of Datong yak. The findings of the semi quantitative PCR displayed that IZUMO1 mRNA expression level was more prominent in testis as compared to other tissues (Figure 1). Based on the difference in the expression pattern between different ages, we hypothesized that the *IZUMO1* expression was higher in 6 years because they coincide with the peak breeding age of yak.

### 3.2. The Expression Level of IZUMO1 mRNA by Quantitative Real Time PCR

The expression level of *IZUMO1* mRNA in different tissues of Datong yak was compared by quantitative real time PCR. The data specified that mRNA expression level of IZUMO1 was significantly (*p* < 0.05) greater in the testis of 6 years, followed by 30 months, 18 months, and 6 months. Moreover, *IZUMO1* mRNA expression level was low in fat, spleen, liver, and heart tissues. However, moderate expression of *IZUMO1* was noted in lungs and kidney tissues (Figure 2).

### 3.3. Characterization of IZUMO1 Gene 

A partial sequence of IZUMO1 gene was cloned and confirmed by colony PCR (Figure 3). Different bioinformatics software was used for the prediction of secondary structures, functional sites, and protein sequence encoded by the *IZUMO1* gene. As shown in Figure 4a, the coding sequence of IZUMO1 protein started with ATG and ended with the TAG stop codon. Furthermore, the coding sequence of the *IZUMO1* gene encoded a protein belonging to the immunoglobulin superfamily consisting of a complete ORF of 1299 bp that encoded 339 amino acids (Figure 4a). The projected secondary structure of the IZUMO1 protein consisted of a helix, extended strand, and coil (Figure 4b). In addition, the protein encoded by yak IZUMO1 contained an immunoglobulin (Ig) domain-associated (Figure 4c). The crystal structure of IZUMO1 was determined by X-ray diffraction 2.50 Å and oligo state is hetero-oligomer (Figure 4d). The multiple alignment amino acid sequence of yak IZUMO1 protein revealed 77.17%, 99.41%, and 94.99% similarity with pig, buffalo, and sheep (Figure 5). Therefore, these results indicated that protein encoded by the *IZUMO1* gene in Datong yak was highly conserved among mammalians.

### 3.4. Western Blotting

The IZUMO1 protein expression in testis was also confirmed by western blot (Figure 6). As we compared the IZUMO1 level in testis of different ages, we evaluated that there was not any significant difference between 30 months and 6 years in the expression of IZUMO1 protein but both these ages showed significantly greater expression of IZUMO1 as compared to 6 months and 18 months ages.

### 3.5. Immunostaining Analysis

Immunostaining was performed further to explore the differences in the morphology of Datong yak testis at different ages (Figure 7). Moreover, immunostaining analysis revealed that IZUMO1 was found in all ages of Datong yak but positive germ cells (red arrow) were exclusively present in the tubules of 30 months and 6 years yak testis compared to 6 months and 18 months. In addition, these observations displayed that IZUMO1 protein expression played a significant role in the development of spermatocytes and its maturation.

## 4. Discussion

IZUMO1 is an important member of the immunoglobulin superfamily (IgSF) and its unique ability to form multimers plays a crucial role in sperm egg fusion and fertilization. Previously, no research was available on IZUMO1 in yak; therefore, we conducted this research on the Datong yak. We cloned the coding region of the *IZUMO1* gene and used different bioinformatics software for the prediction and analysis of protein encoded by the *IZUMO1* gene, with an emphasis on the secondary structure and functional sites. 

The mRNA expression level of *IZUMO1* was significantly greater in the testis of Datong yak as compared to other organs and a high level of *IZUMO1* was observed with the testicular development. High expression of IZUMO1 in the testis and sperm of bovine was analyzed through RT-PCR and western blotting [7]. Rubinstein et al. [16] also reported testis specific expression of *IZUMO1* and the importance of this protein in fertilization was also explored by working on knock-out *IZUMO1* male mice. Fukuda et al. [17] reported that *IZUMO1* mRNA was highly expressed in the testis but not in the epididymis and liver of bulls. Additionally, Kim et al. [18] also showed high expression of the *IZUMO1* gene in the sperm of pigs. Work conducted on sheep and Cashmere goat revealed high expression of IZUMO1 in the testis, which is consistent with our findings. Thus, we concluded that the *IZUMO1* gene might be associated with fertilization in yak.

Our results indicated that protein encoded by the yak *IZUMO1* gene contained a signal peptide, an IZUMO domain, an immunoglobulin (Ig) like domain, and a transmembrane region, which is similar to protein sequence reported in bovine [16]. Similar structure was also reported in mouse [19]. The IZUMO1 cDNAs sequence of cow, sheep, and goat were found to be homologous and at molecular and phylogenetic levels their morphological taxonomy was steady [20]. Predicted protein structure in the present study showed the presence of a helical region in the Ig like domain, which is known to be involved in cell adhesion and might be important for sperm-egg binding [21]. Inoue et al. [22] also speculated that the secondary amino acid structure of IZUMO1 was essential for sperm-egg fusion and suggested that IZUMO1 interacts with the sperm cellular surface at helical regions and promotes sperm and egg membrane fusion. Kim, [7] elucidated that the IZUMO1 is necessary for fusion of oocyte sperm through complex a mechanism of proteins on the sperm and oocyte membrane. Knock-out *IZUMO1* male mice revealed the incapability of sperm to fuse with eggs [10] and showed infertility [23]. The significant role of IZUMO1 in egg sperm fusion mechanism in mammals was also reported [20]. Sebkova et al. [24] stated that IZUMO1 played a major role in acrosome reaction and its presence in the equatorial segment was crucial for sperm-egg fusion. Therefore, the secondary structure of *IZUMO1* might have an important role in the fusion of sperm and egg.

Western blot analysis indicated that the IZUMO1 protein from testis tissue of yak was of 60 kDa. Kim [7] reported that anti-bovine IZUMO1 antibody detected a 65 kDa protein in HEK293 cells which were transfected with bovine IZUMO1. He also reported that 42 kDa protein was monomeric, while 70 kDa proteins was dimeric. In addition, 17 kDa minor form and 45kDa dominant form of bull IZUMO1 were identified in extracts of cauda epididymal spermatozoa [17], 56 KDa protein was observed in wild-type mice [25] and 60 kDa IZUMO1 protein was found in the mammalian sperm [19]. The presence of IZUMO1 in sperms of different species were reported by Gerhardt et al. [12] through western blot analysis, by using anti-porcine IZUMO1 antibody against protein extracts from ejaculated sperm of pig, bovine, and sperm from the cauda pididymal zone of the mouse. Aydin et al. [26] and Ohto et al. [27] explored molecular basis of *IZUMO1* gene in sperm egg recognition, cross species fertilization, and a barrier to polyspermy. They concluded auspicious benefits of the *IZUMO1* gene in the rational improvement of novel non-hormonal contraceptives and treatment of fertility. Our results of qPCR, semi-quantitative PCR, immunostaining and western blotting analysis confirmed that the high expression of the *IZUMO1* gene and its protein in testis of yak and the importance of secondary structure of IZUMO1 protein in the fusion of egg and sperm.

## 5. Conclusions

In conclusion, our results identified the elevated expression of the *IZUMO1* gene with the development of testis and its potential role in the fertilization of Datong yak. Our findings indicated significant expression of the *IZUMO1* gene in testis, while no significant expression was observed in other organs of the yak. The expression of the *IZUMO1* gene was significantly higher in 6 years old yak testis followed by 30, 18, and 6 months old Datong yak testis, which indicated that the expression of the *IZUMO1* gene might be higher during peak breeding age of yak (6 to 7 years). This research revealed the biological role of *IZUMO1* in the reproduction of Datong yak; therefore in the future it could provide a novel prospect to enhance reproduction in yak and provide a concrete basis for the molecular biology underlying fertilization in Yak, as well as in other bovine animals.

## Figures and Tables

**Figure 1 animals-09-00292-f001:**
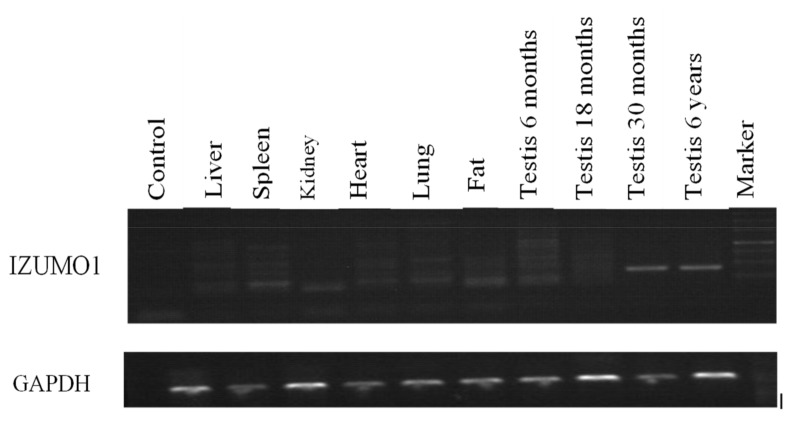
Expression of *IZUMO1* mRNA by semi-quantitative PCR in various tissues of Datong yak.

**Figure 2 animals-09-00292-f002:**
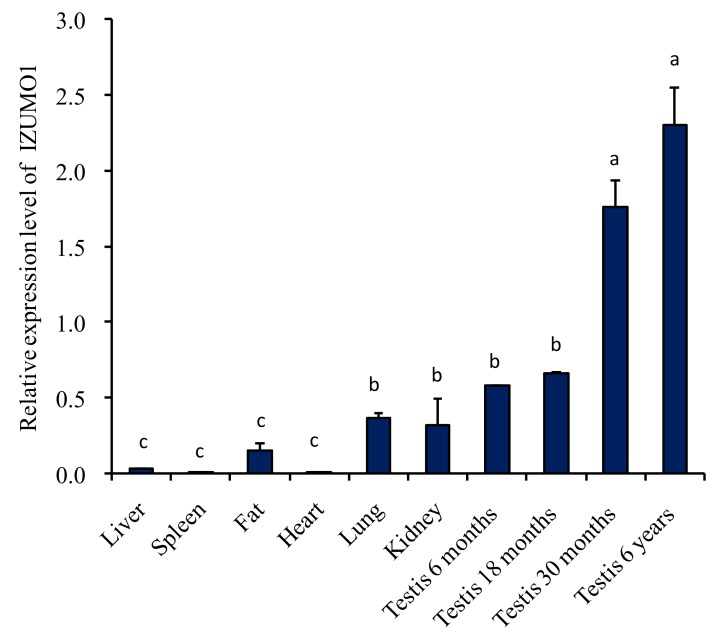
Expression of IZUMO1 mRNA in various organs of Datong yak by quantitative real time PCR. Data are presented as means ± standard deviation (*n* = 4). The error bar indicates the standard deviation (SD). Different letters indicate significant difference (*p* < 0.05).

**Figure 3 animals-09-00292-f003:**
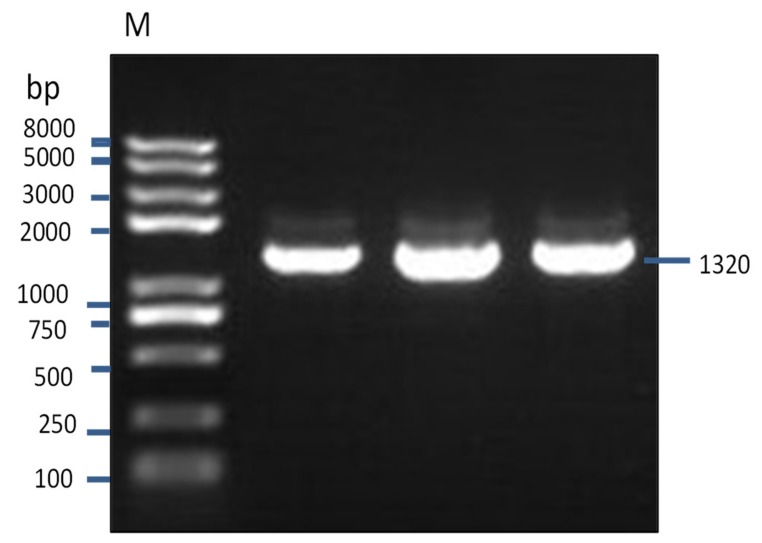
A partial sequence of *IZUMO1* (1320 bp) was amplified from Datong yak testis by colony PCR.

**Figure 4 animals-09-00292-f004:**
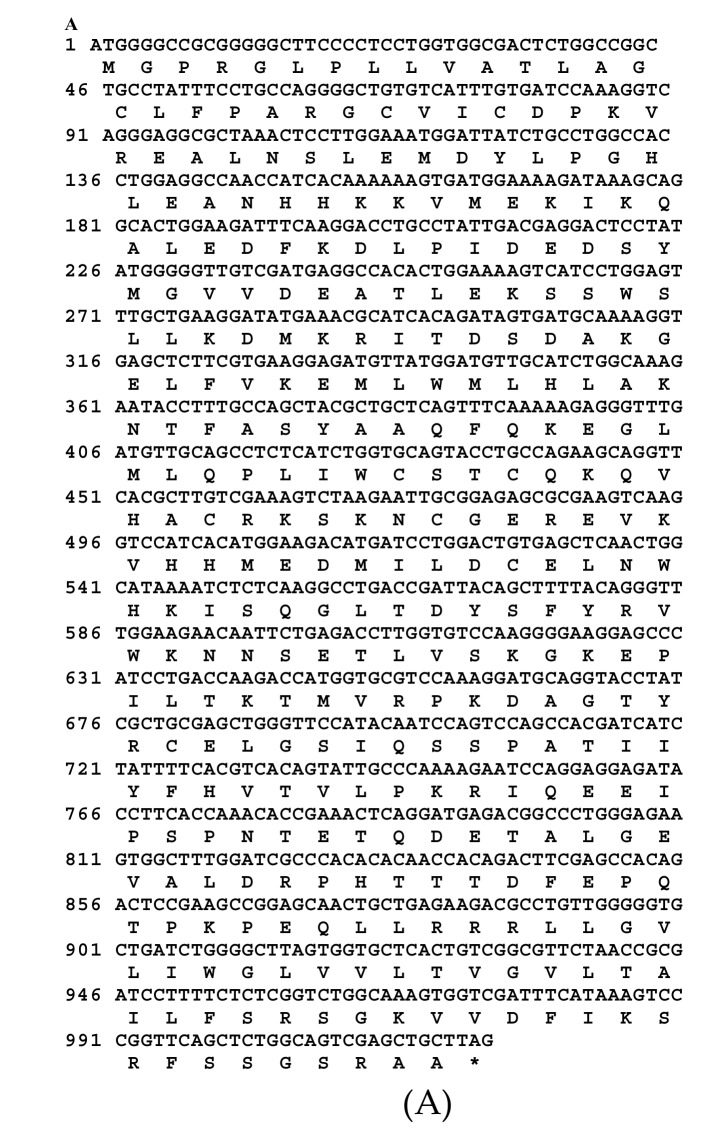
Bioinformatics analysis and prediction of protein sequence encoded by *IZUMO1* from the testis of Datong yak. The protein encoded by yak *IZUMO1* contains an immunoglobulin (Ig) domain-associated. (**A**) Predicted protein and DNA sequences of yak *IZUMO1* gene. (**B**) Predicted secondary structure of *IZUMO1* protein. (**C**) Predicted conserved domain for *IZUMO1* protein. (**D**) Predicted three dimensional structure of protein encoded by *IZUMO1*.

**Figure 5 animals-09-00292-f005:**
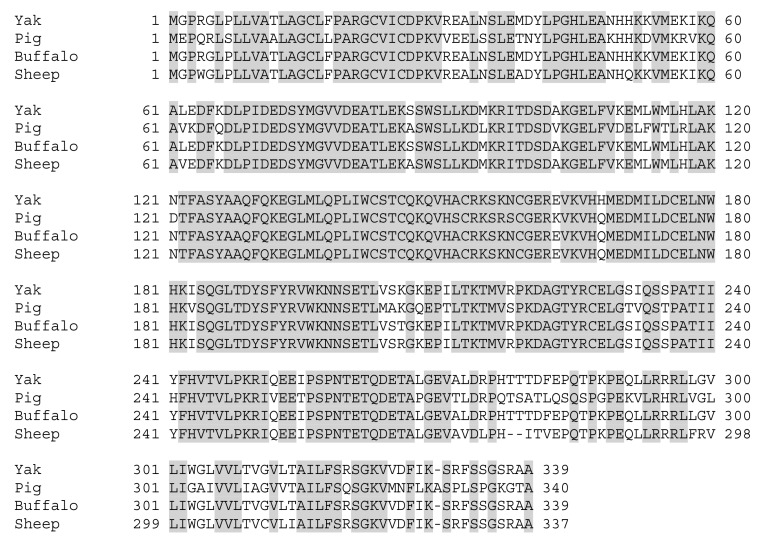
Multiple alignment of amino acid sequences of yak IZUMO1 protein with pig, buffalo, and sheep IZUMO1. Identical amino-acids are represented by shaded boxes.

**Figure 6 animals-09-00292-f006:**
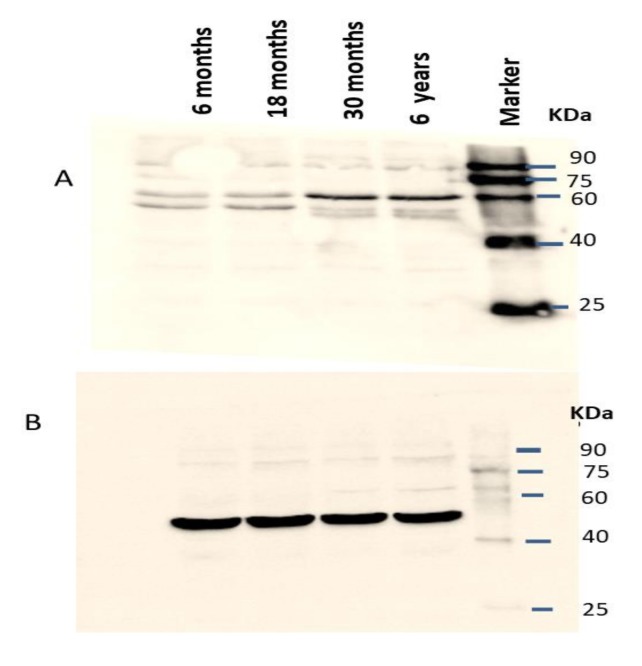
Expression of IZUMO1 protein in Datong yak testis. (**A**) Western blotting for the detection of the expression pattern of IZUMO1 protein in the testis of Datong yak and βeta-actin was used as the internal loading controls (**B**). Relative expression of IZUMO1 protein (**C**).

**Figure 7 animals-09-00292-f007:**
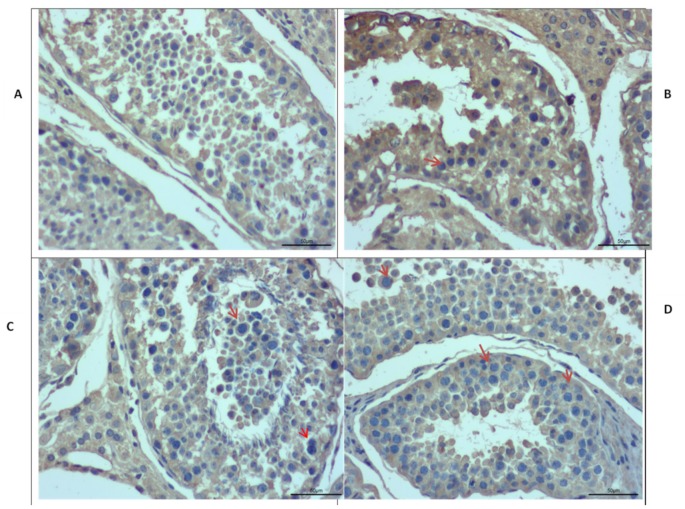
Immunostaining of testis of different ages in Datong yak. Immunostaining analysis indicated that *IZUMO1* is expressed in all ages but positive germ cells (red arrow) are more prominent in the tubules of 30 months and 6 years yak testis compared to 6 months and 18 months. (**A**) 6 months of Datong yak. (**B**) 18 months of Datong yak. (**C**) 30 months of Datong yak. (**D**) 6 years of Datong yak.

**Table 1 animals-09-00292-t001:** Primers used for PCR and qPCR.

Accession No	Gene	Primers Sequence (5′->3′)	Product Length (bp)	Annealing Temperature (°C)
**XM_024979243.1**	IZUMO1 (For cloning)	F: TTCGAGTTGGAAGGACGTGG R: TTCCTTTGAAACCCCCGAGT	1320	59.97 59.15
**XM_024979242.1**	IZUMO1 (For gene expression)	F: GCGTTCTAACCGCGATCCTT R: CTCGACTGCCAGAGCTGAAC	81	60.80 60.74
**NM_001034034.2** **F: AATGAAAGGGCCATCACCATC**	GAPDH	F: AATGAAAGGGCCATCACCAT R: CACCACCCTGTTGCTGTAGCCA	204	55.85 60.00

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
