# Peer review of "Expression Analysis of IZUMO1 Gene during Testicular Development of Datong Yak (Bos Grunniens)"

_animals, 2019, doi:10.3390/ani9060292_

Round 1

Reviewer 1 Report

This is well formulated study but the english grammer and tenses used in this manuscript is not so good. Improve the write up or take help from some english editing services.

Secondly add clearly in the material and methods section that how much total animals were included in this study and how much animals were included in each group and how much study groups were made.

Thirdly, the (Comma sign) should be used in the manuscript  according to the english grammer

Author Response

Response to Reviewer 1 Comments

Reviewer 2 Report

According to the authors this is the first description of the coding regions of IZUMO1 gene and protein in Datong yak, adding knolodge about the possible roles of this protein in yak fertilization.

However, I strongly recommend that, for publication, the discussion must be improved and the english reviewed.

The authors compares their results with others obtained by previous published studies, however their writing is repetitive and some related references are shown in different parts of the discussion (for exemple - text between lines 259 and 270 needs to be improved).

They say that they analysed the IZUMO1 protein with an emphasis on the secondary structures and functional sites, but they present a very shy discussion regarding the role of the functional sites in protein function and stability. This would be a very importante part of the discussion and the most appropriate place to discuss about the protein function, presenting references that brings notion about IZUMO role in egg fertilization, for exemple.

In line 259 the authors said – “Consistent to this research, Zhang et al. [30] revealed the association of IZUMO1 gene with rate of  ovulation, sperm production, and embryonic mortality in yak breeds”. Which one of the results obtained assures that IZUMO1 gene is related with rate of  ovulation, sperm production, and embryonic mortality in yak breed to say that is consistent with Zhang findings?

It might bring some clues, but there is not enough experimental data to afirm that. Which brings to another point, be careful with the conclusions. Again, at the first paragraph of the conclusion the authors afirmed (line 302) – “In conclusion, our results identified the key role of IZUMO1 gene in the development of  testis, spermatogenesis and fertilization of Datong yak”. Its presence in the testis might indicate that, but it does not precisely assure that this is the case.

The figures quality must also be improved and it would be interesting if the authors present an aligment between the yak IZUMO cDNA/protein sequences with the ones from another species. 

Figures 6 says that the IZUMO1 presence is indicated by a brown arrow but the arrows at the figure are red and blue.

Author Response

Response to Reviewer 2 Comments

Round 2

Reviewer 2 Report

The authors have improved the text as suggested and it is apt for publication. However, it is still necessary to revise the English.